# Mesoscopic-scale functional networks in the primate amygdala

**Jeremiah K Morrow[1,2], Michael X Cohen[3,4], Katalin M Gothard[1]***

[1]Department of Physiology, University of Arizona, Tucson, United States; [2]Department of Behavioral Neuroscience, Oregon Health and Sciences University, Portland, United States; [3]Radboud University Medical Center, Nijmegen, Netherlands; [4]Donders Center for Neuroscience, Nijmegen, Netherlands

**Abstract** The primate amygdala performs multiple functions that may be related to the anatomical heterogeneity of its nuclei. Individual neurons with stimulus- and task-specific responses are not clustered in any of the nuclei, suggesting that single-units may be too-fine grained to shed light on the mesoscale organization of the amygdala. We have extracted from local field potentials recorded simultaneously from multiple locations within the primate (*Macaca mulatta*) amygdala spatially defined and statistically separable responses to visual, tactile, and auditory stimuli. A generalized eigendecomposition-based method of source separation isolated coactivity patterns, or components, that in neurophysiological terms correspond to putative subnetworks. Some component spatial patterns mapped onto the anatomical organization of the amygdala, while other components reflected integration across nuclei. These components differentiated between visual, tactile, and auditory stimuli suggesting the presence of functionally distinct parallel subnetworks.

## Introduction

The division of the amygdala into nuclei reflects the developmental origins and the input-output connections of each nucleus with other structures (*Amaral et al., 1992*; *Pessoa et al., 2019*; *Swanson and Petrovich, 1998*). Cell-type-specific circuit dissection of the rodent amygdala revealed further compartmentalization of the amygdala (reviewed by *Duvarci and Pare, 2014*; *Fadok et al., 2018*; *Gafford and Ressler, 2016*; *Janak and Tye, 2015*). However, the anatomical compartmentalization is rarely reproduced by the response properties of single neurons (e.g., *Beyeler et al., 2018*; *Kyriazi et al., 2018*; *Morrow et al., 2019*; *Putnam and Gothard, 2019*). This is not surprising given the difficulty in capturing the activity of functional networks by subsampling the constituent neurons, especially when the network includes neurons with multidimensional response properties (*Gothard, 2020*). At the other end of the spectrum of scale, neuroimaging techniques that monitor the activity of brain-wide networks often lack sufficient resolution to capture nuclear or subnuclear activation in the amygdala (*Bickart et al., 2012*; *Roy et al., 2009*).

The local field potentials (LFPs), reflecting the aggregate activity of hundreds to tens of thousands of neurons (*Buzsáki et al., 2012*; *Einevoll et al., 2013*; *Pesaran et al., 2018*), may provide a more adequate mesoscopic-scale view of intra-amygdala activity. Although task-relevant LFP signals have been successfully extracted from the amygdala of non-primate species in the context of fear conditioning (*Courtin et al., 2014*; *Paré et al., 2002*; *Seidenbecher et al., 2003*; *Stujenske et al., 2014*), anticipatory anxiety (*Paré et al., 2002*), and reward learning (*Popescu et al., 2009*), remarkably little is known about the information content of the local field potentials in the primate amygdala.

We recently reported that the majority of multisensory neurons in the monkey amygdala not only respond to visual, tactile, and auditory stimuli, but also discriminate, via different spike train metrics, between sensory modalities and even between individual stimuli of the same sensory modality (*Morrow et al., 2019*). Multisensory neurons and neurons selective for a particular sensory modality

***For correspondence:**
kgothard@email.arizona.edu

**Competing interests:** The authors declare that no competing interests exist.

were not clustered in any nucleus or subnuclear region, suggesting an organization scheme in spatially distributed but functionally coordinated networks.

We expected that the combined excitatory, inhibitory, and neuromodulatory effects elicited by different sensory modalities would be better captured in the dynamics of the LFPs than by the response properties of single neurons (*Buzsáki et al., 2012*; *Mazzoni et al., 2008*). Specifically, we explored the hypothesis that stimuli of different sensory modalities elicit different spatiotemporal patterns of activity in the local field potentials recorded from linear arrays of electrodes. Rather than analyze the signals on each contact independently, we used a method of non-orthogonal covariance matrix decomposition called generalized eigendecomposition (GED) to identify co-activity patterns across the set of contacts. Because neural sources project linearly and simultaneously to multiple contacts, linear multivariate decomposition methods are highly successful at separating functionally distinct but spatially overlapping sources (*Parra et al., 2005*). These co-activity patterns (also called 'components') are the product of putative functional subnetworks within the brain. GED has been shown to reliably reconstruct network-level LFP dynamics in both simulated and empirical data that are often missed by conventional methods of source-separation (*Cohen, 2017a*; *de Cheveigné and Parra, 2014*; *Parra et al., 2005*; *Van Veen et al., 1997*). Here, we adapted this method to our data and discovered multiple, statistically dissociable patterns of subnetwork activity tied to the functional and spatial structure of the amygdala that was missed by the analysis of single-unit activity (*Morrow et al., 2019*).

## Results

We monitored LFPs simultaneously from the entire dorso-ventral expanse of the amygdala using linear electrode arrays (V-probes, Plexon Inc, Dallas, TX) that have 16 equidistant contacts (400 µm spacing, *Figure 1*). We systematically sampled the medio-lateral and anterior-posterior regions of the amygdala during different recording sessions. In each session the signals were referenced to the common average across the 16 contacts; this ensured that our signals were locally generated and not volume conducted from distant regions.

Sets of eight visual, tactile, and auditory stimuli were presented to the monkey as static images, gentle airflow, and random sounds (*Figure 1*). Each stimulus was presented 12–20 times and was followed by juice reward. All stimuli were chosen to be unfamiliar and devoid of any inherent significance to the animal (assuming that images of fractals or sounds made by musical instruments that were not previously encountered by the monkeys have no intrinsic value). Given that the delivery of all stimuli was followed by the same reward, the learned significance of these stimuli was not different across sensory modalities. Stimuli with socially salient content like faces or vocalizations were avoided, as were images or sounds associated with food (e.g., pictures of fruit or the sound of the feed bin opening). Airflow nozzles were never directed toward the eyes or into the ears to avoid potentially aversive stimulation of these sensitive areas.

The LFP signal was compared between a baseline window (from −1.5 to −1.0 s relative to fixation cue onset) and a stimulus window (from 0 to +1.0 s relative to stimulus onset). Ninety-five percent of the contacts showed significant changes in LFP activity relative to baseline (623/656, 95.0% of all contacts, Wilcoxon rank-sum tests; Bonferroni corrected for 16 comparisons, $\alpha$ = 0.01/16 = 0.000625). Rather than analyze the signals on each contact individually, we used a guided source-separation method, called generalized eigendecomposition (GED), to identify covariation patterns across the contacts that were maximally activated during stimulus delivery relative to the baseline period (*Figure 2a–c*). We then assessed whether these covariation patterns, also called 'components,' were spatially defined and modality-specific (*Figure 2b–g*) (for further details see Materials and methods). Note that unlike principal components analysis, GED components do not have an orthogonality constraint, which facilitates physiologically interpretability (*Cohen, 2017a*).

We found between 1 and 5 simultaneously active and statistically significant components during each recording session (*Figure 2h–i*), leading to a total of 116 components obtained from 41 sessions (significance was computed via permutation testing with corrections for multiple comparisons; see Methods). In some of the recording sessions a subset of contacts on the linear array were estimated to be located outside of the amygdala. Removing these contacts from the analysis eliminated 24 components (i.e., 92 components were attributable solely to activity recorded in the amygdala), emphasizing the importance of the spatial location of each contact (*Figure 2j–k*). Only results from

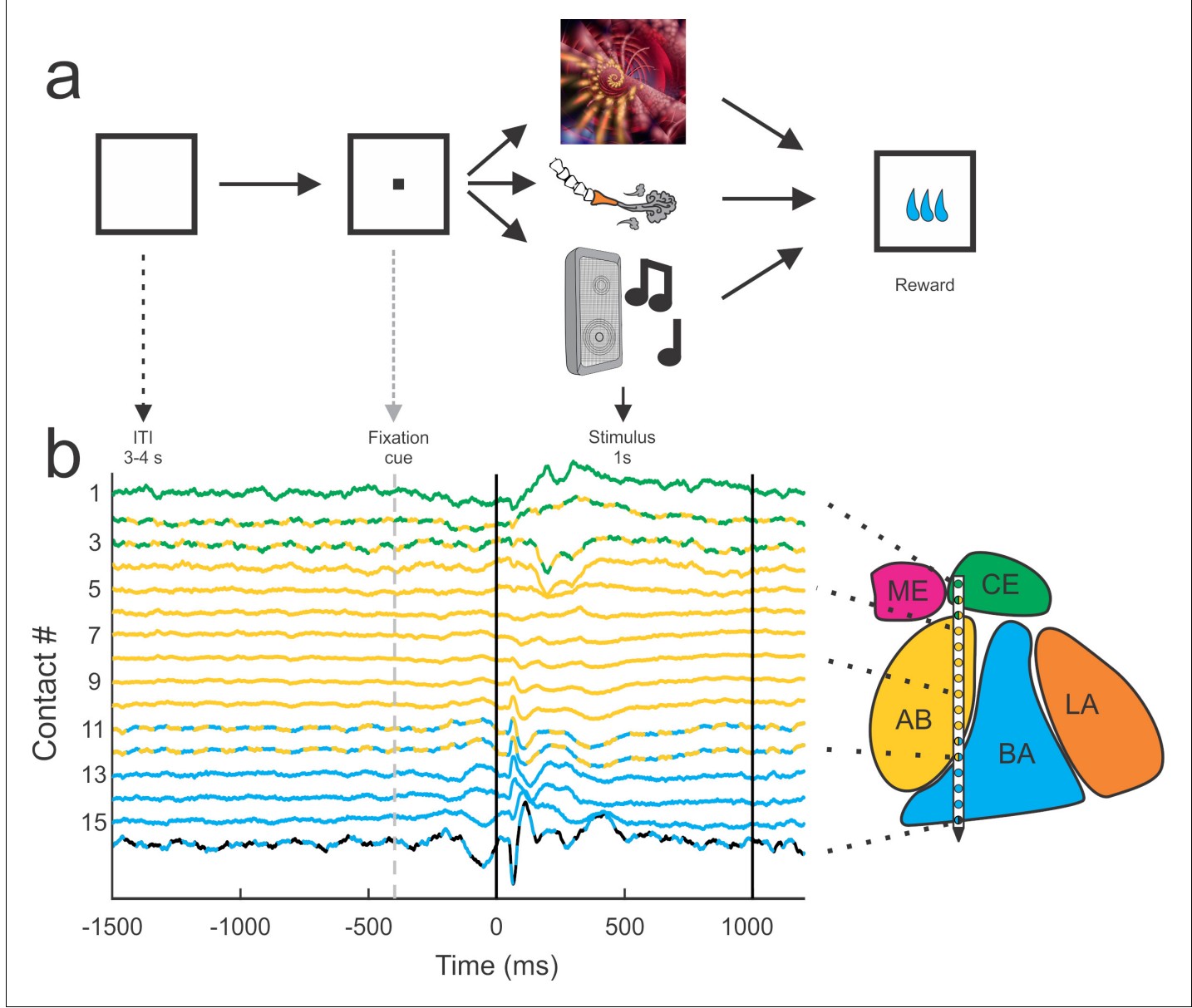

**Figure 1.** Behavioral setup and example average traces across all trials on each of the 16 contacts. All trials were separated by a 3–4 s inter-trial interval (ITI) and preceded by a visual fixation cue presented at the center of a monitor in front of the animal. If the monkey successfully fixated on the cue, the cue was removed and a single stimulus from one of the three sensory modalities (visual, tactile, auditory) was presented. All trials were followed by a brief delay (0.7–1.4 s) before a small consistent amount of juice was delivered. The color code of the peri-event LFP activity corresponds to the estimated location of the recording electrode within the nuclei of the amygdala. The lines with alternating colors refer to contacts located within 200 μm of an anatomical boundary between nuclei. ME = medial, magenta; CE = central, green; AB = accessory basal, yellow; BA = basal, cyan; LA = lateral, orange; non-amygdala contacts are colored black.

significant components are discussed for the remainder of this manuscript (p<0.05 corrected for multiple comparisons via permutation testing; details in Materials and methods).

The relative contribution of the signal from each contact to a particular component can be visualized in a 'component map' (*Figure 2d and f*). This map can then be co-registered with the anatomical map of the amygdala to reveal whether contacts in different nuclei contribute to the components in distinct ways. Each component also has its own time series (*Figure 2e and g*) that results from passing the raw LFP signals though the associated eigenvector to obtain a reconstructed signal that

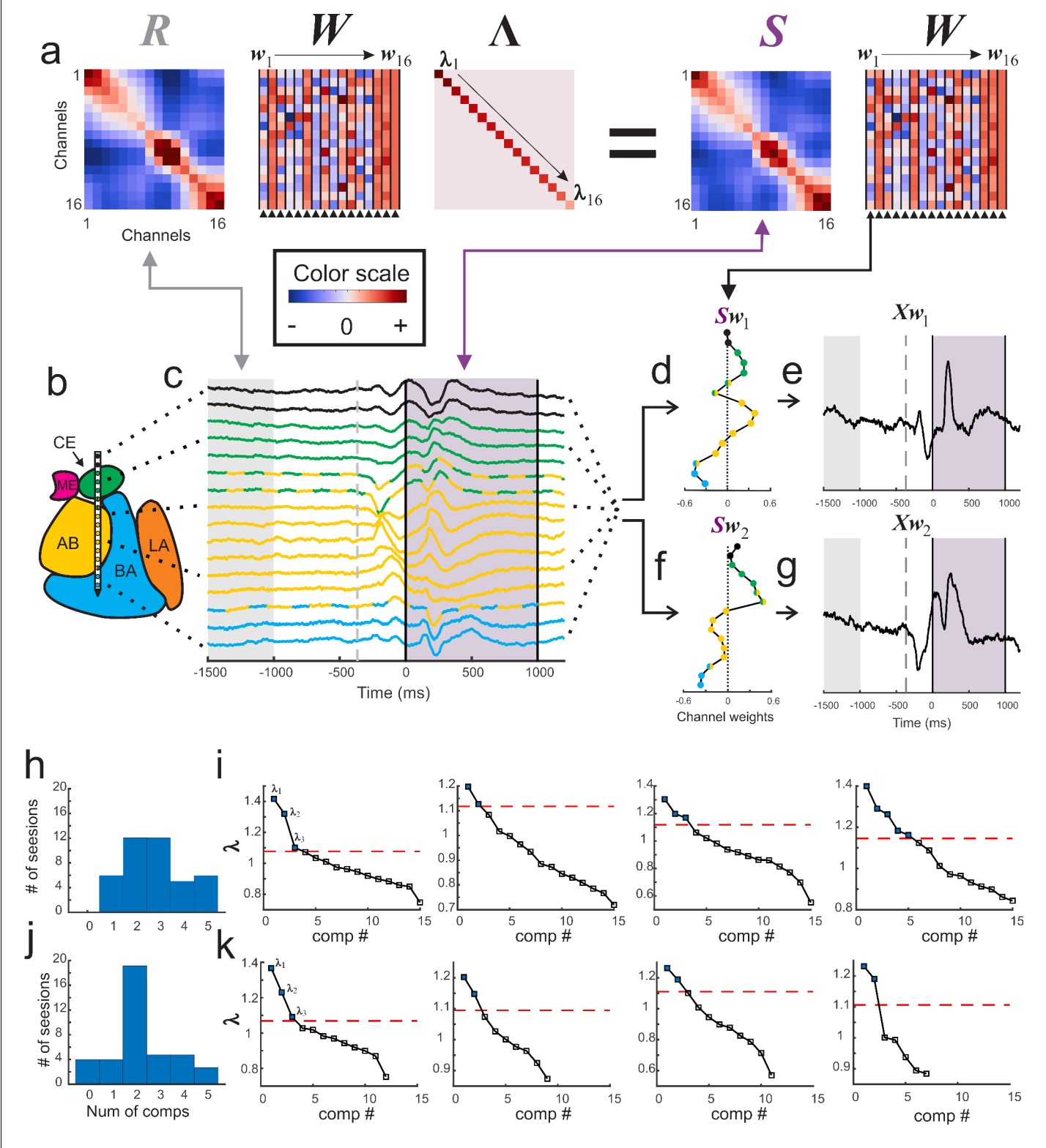

**Figure 2.** Visualization of GED. (a) The elements of the equation for generalized eigendecomposition (RWΛ = SW), where R and S are 16x16 covariance matrices (corresponding to the 16 contacts) derived from baseline and stimulus activity respectively. These contact covariance matrices were generated for each trial and then averaged over trials (see Materials and methods). The columns in the W matrix (highlighted by the arrowheads below the plot) contain the eigenvectors. Each eigenvector from the matrix can be written as $w_n$ (indicated by the $w_1 \rightarrow w_{16}$ above the plots). The diagonal matrix of eigenvalues is represented by Λ. The individual eigenvalues are denoted by $\lambda_1 \rightarrow \lambda_{16}$ shown along the diagonal. A general color scale for the

*Figure 2 continued on next page*

*Figure 2 continued*

heatmap images is shown under the left side of the equation. (**b**) MRI-based reconstruction of recording sites in this session. (**c**) The average peri-stimulus LFP for each contact (color convention is the same as in *Figure 1*). Light gray shading represents the baseline period and the gray dotted line denotes fixspot onset (note that these data are aligned to stimulus onset; the baseline and fixspot indicators shown here are for illustrative purposes only). The purple shading marks stimulus delivery. Note that the data in (**c**) are contained in a time-by-contacts matrix (represented by X in subsequent equations). (**d**) Example of the component map associated with the largest eigenvalue (created by multiplying the stimulus covariance matrix with an eigenvector and termed $Sw_1$). (**e**) The component time series associated with this eigenvalue. This time series (represented as $Xw_1$) is created by multiplying the LFP data matrix (i.e., X) with the eigenvector in the first column of the (W) matrix ($w_1$). (**f, g**) the same as for (**d**) and (**e**) but associated with the second largest eigenvalue. (**h**) Histogram of the number of significant components per session when including all contacts regardless of location (i.e., within or outside of the amygdala). (**i**) Scree plots of the eigenvalues derived from GED from four example sessions when all contacts were included in analysis regardless of location. In the scree plots, the points above the dotted line correspond to the significant eigenvalues. (**j**) Histogram of the number of significant components per session when only amygdala contacts were included in the analysis. (**k**) Scree plots of the eigenvalues from the same sessions in (**i**) but only using amygdala contacts. Note that decreasing the number of contacts decreases the number of total components that GED is able to extract. This does not always result in decreases in the number of significant components (left two panels); however, small (right middle) and occasionally large (far right) decreases in the number of significant components were observed when removing non-amygdala contacts from the analysis.

corresponds to the estimated output of the subnetwork captured in the component (*Haufe et al., 2014b*).

## GED components map onto anatomical boundaries

The component maps show the extent to which the signals recorded from each anatomically local-ized contact contribute to the putative subnetwork captured by the component (*Figure 3*). The fur-ther a contact weight value is from 0, the more the signal on that particular contact contributes to the component/subnetwork. Adjacent contacts that fall on one side of the zero line co-vary, and thus contribute similarly to the same component. Values of opposing signs (positive vs negative) make opposing contributions to the components (i.e., signals on contacts associated with positive weights inversely covary with the signals on contacts associated with negative weights). We applied a change-point detection algorithm (*Killick et al., 2012*; *Lavielle, 2005*) to cluster the contacts according to shifts in the rolling average of adjacent values in the component maps (see Materials and methods). The boundaries of the statistical clusters matched both internal and external anatomical boundaries of the amygdala (*Figure 3*), which were estimated through a combination of high-resolution MRI and histology (*Figure 3—figure supplement 1*). This is important because GED is a purely statistical decomposition that is not constrained by anatomical information, spatial arrangement, or relative distances of the electrode positions. Moreover, this method discovered subnuclear divisions that neuroimaging would not be able to detect but are well known from histo-logical analyses. For example, in *Figure 3e and f* the horizontal dotted lines intersecting the basal nucleus correspond to known boundaries between the magnocellular, intermediate, and parvocellu-lar subdivisions (*Amaral et al., 1992*).

We quantified the match between the statistical grouping of contacts and the anatomical group-ing of contacts as a percent overlap (see Methods). The overall number of matching contacts was 723/1072 (67.4%). We then determined which of the components (rank-ordered first, second, etc., see histogram in *Figure 2h–j*) showed the best match between the statistical grouping and the ana-tomical grouping of contacts. Only the first components showed statistically better than chance overlap (249/349, 71.3%, paired t-test, t = 2.76, df = 40, p=0.009, 1000 permutations, Bonferroni corrected for five comparisons, $\alpha = 0.05/5 = 0.01$).

## GED components discriminate between sensory modalities

It is possible that the selectivity for sensory modality seen at the single-unit level in the amygdala (*Morrow et al., 2019*) is also evident in the components obtained by GED. Time-frequency analyses of the component time series showed that each subnetwork responded selectively to visual, tactile, or auditory stimuli or a combination thereof (*Figure 4*). Of the 116 identified components, 102 dis-criminated between sensory modalities in at least one frequency range for at least one sensory modality (cluster-mass t-tests, $\alpha = 0.01$, see Methods), as shown by the change in power elicited by stimuli of each sensory modality (*Figure 4a–f*). Only 13 out of these 102 components showed

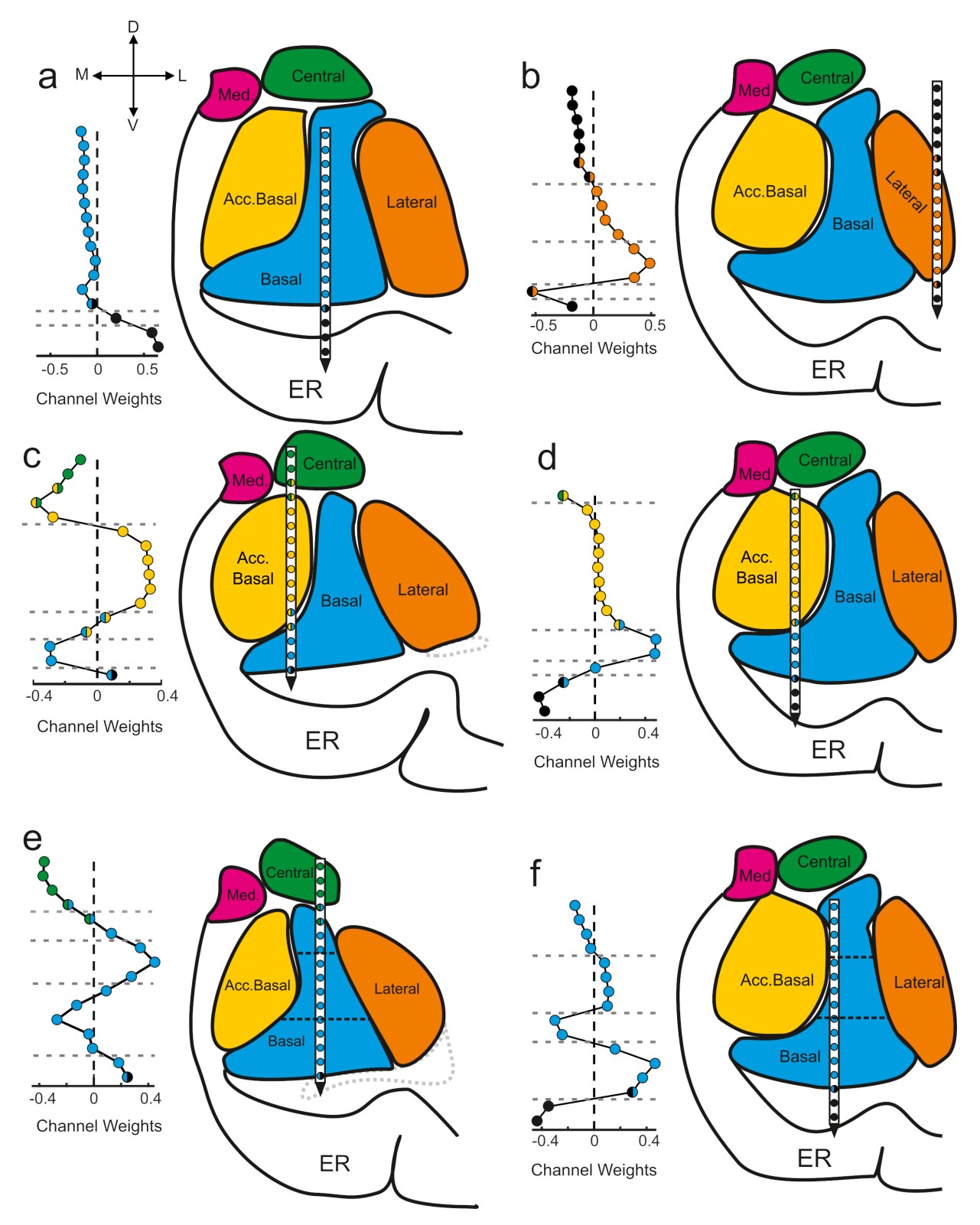

**Figure 3.** Components map onto anatomical boundaries. Six examples of how component maps (left) match the MRI-based reconstruction of the electrode array positions (right). On each component map the x axis corresponds to the weight calculated for each contact (i.e., see Sw in *Figure 2*) of the V-probe and the y axis lists the contacts. In each panel, the colors of the contact weights match their estimated nuclear locations following the same convention as in previous figures. Gray dotted lines in the component map plots denote change points in the contact weights that statistically

*Figure 3 continued on next page*

*Figure 3 continued*

separate groups of contacts based on their coactivity patterns (see Methods). Ventricles are contoured in light gray dotted lines. Med = medial nucleus; Acc. basal = accessory basal nucleus; ER = entorhinal cortex. (a–b) Examples of large transitions in contact weights at boundaries between the amygdala and surrounding tissue, particularly on the ventral contacts. (c–d) Examples of component maps in which the statistical clustering (dotted horizontal lines) match the geometry of the nuclear boundaries. (e–f) Two examples of component maps in which the statistical boundaries correspond to known subnuclear divisions of the basal nucleus.

The online version of this article includes the following figure supplement(s) for figure 3:

**Figure supplement 1.** Histological verification of electrode placement.

responses restricted to a single sensory modality. These results replicate, at a larger spatial scale, our findings from the single unit analyses (*Morrow et al., 2019*), that is, that the majority of sensory signals processed in the amygdala are multisensory.

The power profiles of the component time series were modality-selective but not modality specific, that is, the processing of visual, tactile, or auditory stimuli was not restricted to a particular frequency domain or a particular latency relative to stimulus onset. Sensory modalities were better differentiated at lower frequencies (Wilcoxon rank-sum tests, $\alpha = 0.01$; *Figure 4g*). Typically, the largest changes in low-frequency power (<10 Hz) were elicited by visual stimuli, followed by tactile stimuli, with the lowest values for auditory stimuli (*Figure 4g*), confirming in physiological terms the richer anatomical connectivity of the monkey amygdala with visual areas compared to auditory or tactile areas (*Amaral et al., 1992*). The same observation holds when only assessing components with multisensory responses (72/89 components) (Wilcoxon rank-sum tests, $\alpha = 0.01$; *Figure 4h–j*); visual stimuli elicited larger power changes than tactile, which in turn elicited larger changes than auditory stimuli.

We created a spectral profile based on the significant component signals across all datasets (*Figure 4k*). We then used principal components analysis to extract the features of the component activity that were most prominent across recording sessions. While the spectral profile from the first component roughly followed the 1/f function typically observed in LFP signals, the subsequent spectra showed peaks around the delta (3.5–4.5 Hz), theta (6-8), low-gamma (~38 Hz), and high-gamma (63–75 Hz) frequency bands. In order to determine whether these spectral profiles were truly amygdala generated or were influenced by non-amygdala sources, we re-created the spectra using the components that included only electrodes estimated to be within amygdala boundaries (based on MRI co-registration). While some changes in the spectra were expected from the removal of non-amygdala contacts, it is noteworthy that the prominent spectral peaks (for example, at 4, 7, 38, and 70 Hz) were preserved (*Figure 4l*). The preservation of these spectral peaks after removing possible influence of non-amygdala electrodes shows that the primate amygdala exhibits prominent rhythms at peaks often observed in the cortex. Examples of how removing non-amygdala contacts from the analyses impacts the components in individual recording sessions are shown in *Figure 4—figure supplement 1*.

## Modality-specific GED analyses

Given the activity modulation by sensory modality (*Figure 4g–j*), we re-computed the GED separately for each modality to investigate whether the components could have been driven by a single sensory modality (see Methods). No significant differences were observed between the number of components (visual = 81, tactile = 79, and auditory = 71) extracted via modality-specific GED ($\chi^2$ = 0.37, p=0.83) (*Figure 5a–c*). Furthermore, the sensory-modality-specific component maps were significantly correlated with the component maps that combined all three sensory modalities (*Figure 5c*). We found, however, that the component maps based solely on visual trials matched better the nuclear boundaries than the component maps based only on either auditory or tactile trials (*Table 1*). Specifically, the changepoint positions in the first and second visual component maps coincided with the estimated anatomical boundaries of the nuclei (paired t-tests, p=0.0006 and p=0.001, respectively; same parameters as previous analyses, see *Table 1* for full stats). Overall, these results suggest that the main GED findings reflect a mixture of sensory-independent and sensory-dependent functional organization.

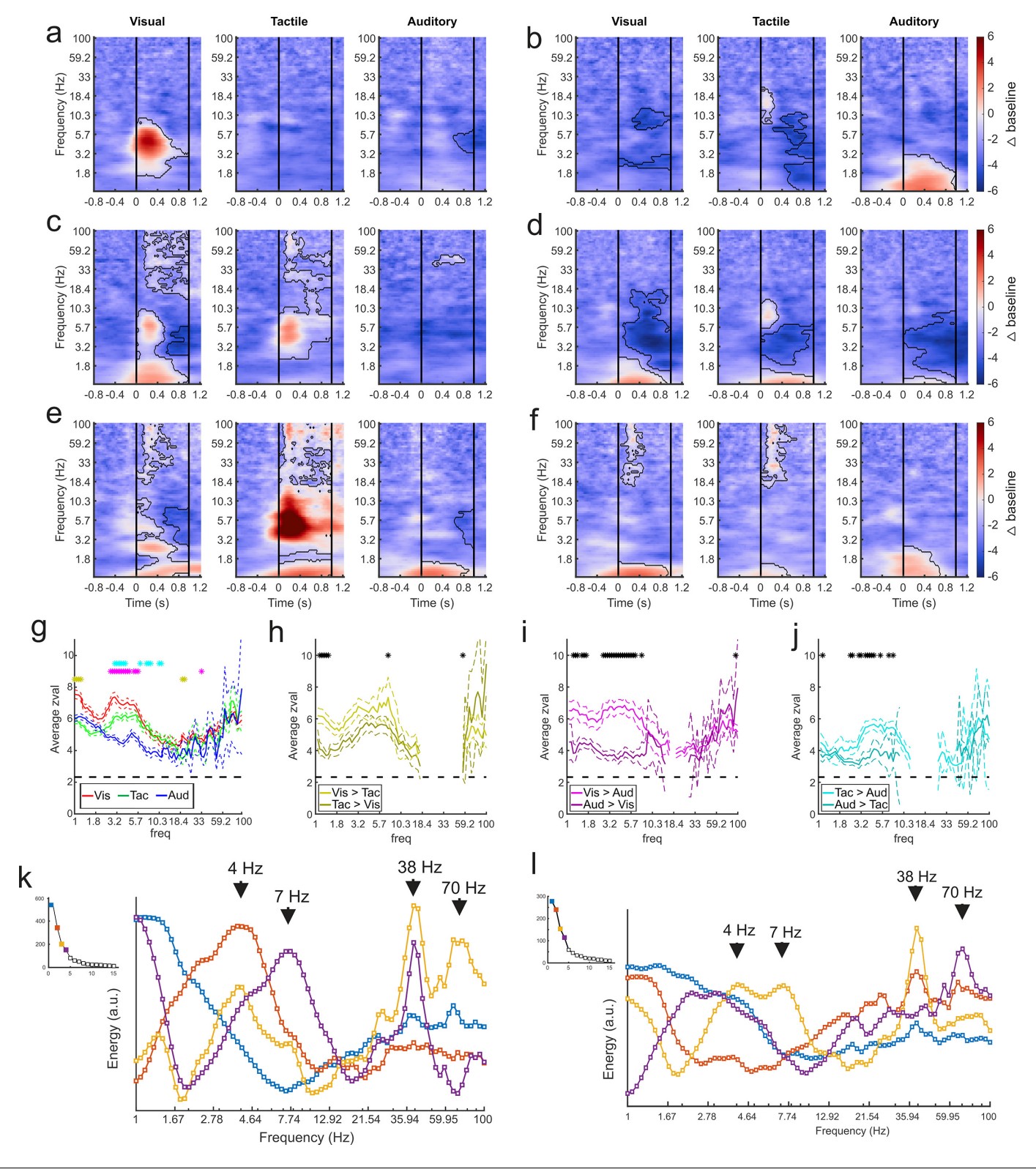

**Figure 4.** GED-based components show modality-selective but not modality-specific changes in multiple frequency bands. (**a–f**) Example time-frequency plots created from the GED-based components separated by stimulus modality (i.e., visual, tactile, and auditory). Each time-frequency plot shows the relative difference in power between the baseline and stimulus periods for a GED-based component time series (scale at far right, 1–100 Hz, logarithmic scale). Black contoured lines denote clusters of significant changes in power between baseline and stimulus delivery (see

*Figure 4 continued on next page*

*Figure 4 continued*

Materials and methods). Solid lines at 0 and 1 s denote the start and end of stimulus delivery. (**a**) Increase in power centered around 4.5 Hz elicited by visual stimuli. (**b**) Increase in power across low (1–3 Hz) frequencies elicited by auditory stimuli with minor power changes for visual and tactile stimuli. (**c**) Moderate increase in power from 1 to 2 and 5–8 Hz for visual stimuli and 4–8 Hz for tactile stimuli. (**d**) Moderate decreases in power for all stimuli, centered around 4 Hz. (**e**) Disparate responses to visual, tactile, and auditory stimuli. Note how multiple frequencies bands show time varying increases in power for all three stimuli, however, the time-frequency power around 1 Hz is similar between the sensory modalities. (**f**) A component with fairly similar responses across sensory modalities but note the lack of high frequency activity for auditory stimuli. (**g**) Maximum separation of the selectivity for sensory modality occurs below 10 Hz. Solid lines represent the mean z-value within significant clusters at each frequency for each modality (visual = red, tactile = green, auditory = blue). Dotted lines denote mean +/- 5 standard errors of the mean. Gold asterisks denote significant differences between the visual and tactile responses, magenta asterisks denote significant differences between visual and auditory, and cyan asterisks denote differences between visual and auditory (Wilcoxon rank-sum tests, p<0.01). (**h–j**) Pairwise comparisons of selectivity between modalities for components that responded to multiple sensory modalities. Solid lines represent mean and dotted lines represent the mean +/- 5 standard errors of mean. Black asterisks denote significant differences (Wilcoxon rank-sum tests, p<0.01). (**k**) Spectral profiles were generated from the significant components from all sessions. PCA was then used to extract the prominent features of these profiles (right). Corresponding scree plots shown on the left (see Materials and methods). These plots were created using all available data regardless of estimated location of the contact. The spectral profile and scree plots were color coded according to the rank of the associated component (1st = blue; 2nd = orange; 3rd = yellow; 4th = purple). The power at each frequency is plotted in arbitrary units of energy. Arrows highlight peaks in the spectra that correspond to frequencies that have been extensively studied in other brain regions (i.e., delta, theta, and low and high gamma). (**l**) Same as in (**k**) but only using data from contacts that were estimated to be within the amygdala. *Figure 4—figure supplement 1* shows how removing non-amygdala contacts can – but does not necessarily – impact spectral profiles in two individual recording sessions.

The online version of this article includes the following figure supplement(s) for figure 4:

**Figure supplement 1.** Effects of removing non-amygdala contacts from the GED-based analyses.

## Discussion

Neither the fine-grained single unit literature nor the coarser neuroimaging literature brought conclusive evidence for or against the functional compartmentalization of the primate amygdala. Only a few single unit studies reported an uneven distribution of neurons with particular response properties across the nuclei (e.g., *Grabenhorst et al., 2019*, *Grabenhorst et al., 2016*; *Mosher et al., 2010*; *Zhang et al., 2013*). To complicate things further, an increasing number of single unit studies reported that neurons in the amygdala respond to multiple stimulus dimensions and task parameters (*Kyriazi et al., 2018*; *Morrow et al., 2019*; *Munuera et al., 2018*; *Putnam and Gothard, 2019*; *Saez et al., 2015*; *Gothard, 2020*). These multidimensional neurons are not clustered in any nuclei or subnuclear region, which seems to be at odds with the compartmentalized view of the amygdala. Local field potentials, multi-unit activity, or higher yield single-unit recordings could provide better approximations of network-level activity and may therefore help to resolve this apparent contradiction (for a critique of neuron-based concepts see *Buzsáki, 2010*; *Yuste, 2015*).

Remarkably little is known about the neurophysiology of the primate amygdala at the network level, due in part to its layerless architecture that is not expected to generate the predictable LFP patterns observed in layered structures like the neocortex or the hippocampus. In the primate amygdala, each nucleus and nuclear subregion has different cytoarchitecture and also distinct input-output connections with both cortical and subcortical structures (*Amaral et al., 1992*; *Ghashghaei and Barbas, 2002*; *Sah et al., 2003*, *Pessoa et al., 2019*). For example, projections from the amygdala to dopaminergic neurons in the substantia nigra mainly arise from the central nucleus. These central nucleus projections are further distinguished into medial and lateral subdivisions that terminate in separate subregions of the substantia nigra (*Fudge and Haber, 2000*). Likewise, the subregions of the dorsolateral prefrontal cortex are reciprocally connected to distinct nuclear subdivisions of the basolateral complex (*Amaral and Price, 1984*; *Ghashghaei and Barbas, 2002*). This organization could give rise to parallel, nucleus-anchored processing loops that induce nuclear-specific patterns of neural activity. It is also possible that the rich intra-amygdala connectivity (*Bonda, 2000*; *Pitkänen and Amaral, 1991*) distributes signals arriving to or originating from a particular area of the amygdala across multiple nuclei, evening out functional differences expected based on structural consideration alone. The human neuroimaging literature already alluded to this type of organization, based on anatomical connectivity (*Bickart et al., 2012*) or resting state fMRI data (*Roy et al., 2009*).

Here, we provide evidence for a mesoscopic organization of the amygdala, into putative subnetworks that often respect nuclear boundaries, but in some cases reflect synchronization of neural

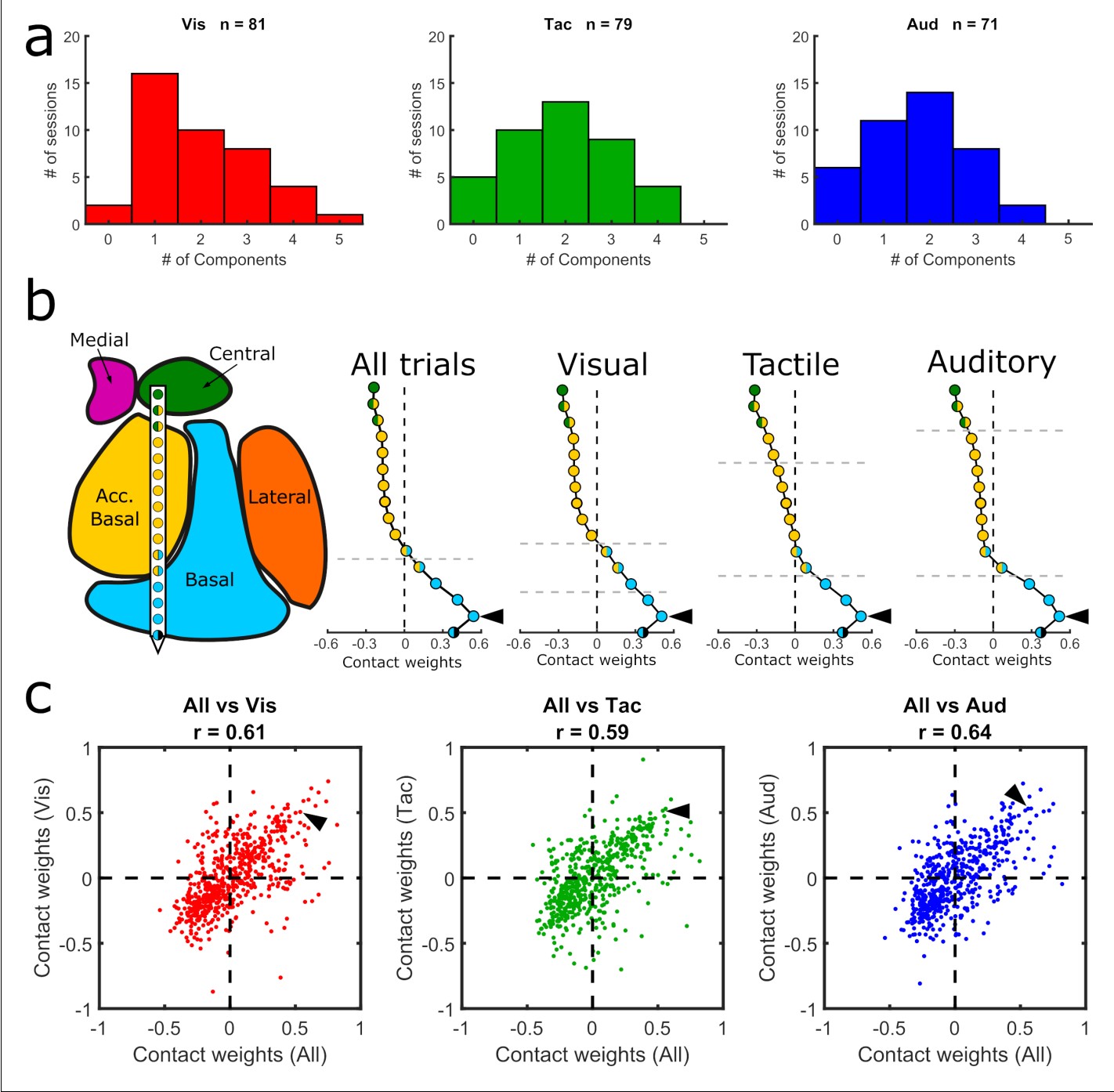

**Figure 5.** Comparison of GED components across sensory modalities. (a) Histograms showing the distribution of significant components from visual, tactile, and auditory trials. (b) MRI-based reconstruction of recording sites from an example session (left) and component maps (right) generated from data collected in the same example session using all trials (left) and only visual, tactile, or auditory trials (last three plots in the row). All component maps shown in panel b are based on the first component resulting from each analysis. (c) Pairwise correlations between component maps generated using all trials (x-axis) vs. component maps generated from modality-specific components (y-axis). Each dot shows the paired map projections onto one electrode from one component. Arrowheads in (b) show how the value from a single contact in each of the component maps projected into the scatter plots in (c).

**Table 1.** Only visual components significantly map onto anatomical boundaries.

Only the first and second visual component maps matched the anatomical boundaries when correcting for multiple comparisons (paired t-tests, Bonferroni corrected for 12 comparisons, p=0.05/12 = 0.004). C.I. indicates the 95% confidence interval.

|  | Component #1 | Component #2 | Component #3 | Component #4 |
|---|---|---|---|---|
| Visual | t(38)=3.75 p=0.00059 C.I. = [0.4, 1.56] | t(22)=3.74 p=0.0011 C.I. = [0.57, 1.98] | t(12)=0.25 p=0.81 C.I. = [−0.54, 0.68] | t(4)=−0.054 p=0.62 C.I. = [−1.80, 1.22] |
| Tactile | t(35)=0.63 p=0.53 C.I. = [−0.34, 0.65] | t(25)=1.97 p=0.06 C.I. = [−0.02, 0.97] | t(12)=1.42 p=0.18 C.I. = [−0.27, 1.29] | t(3)=1.87 p=0.16 C.I. = [−1.03, 3.97] |
| Auditory | t(34)=−2.46 p=0.019 C.I. = [−0.76,−0.07] | t(23)=−0.74 p=0.47 C.I. = [−0.58, 0.27] | t(9)=−0.02 p=0.99 C.I. = [−0.96, 0.94] | t(1)=−0.97 p=0.51 C.I. = [−14.15, 12.14] |

populations across multiple subnuclear regions, that were identified by guided source separation of the local field potentials recorded from multi-contact electrodes. The presence of *any* identifiable subnetworks demonstrates that a layered architecture is not a prerequisite for meaningful organization of LFP activity. The presence of *multiple* subnetworks suggests functional compartmentalization. Importantly, the strongest of the subnetworks were contoured by the internal and external boundaries of the nuclei of the amygdala. This is remarkable because GED is 'blind' to the spatial arrangement of the contacts on the V-probe and to the location of the V-probe in the amygdala. This is evidence that a mesoscale physiological feature of the amygdala (the LFP), unlike the single units, is bound by anatomical constraints. The implications of this finding are twofold: (1) expanding GED to three-dimensions (using multiple linear electrode arrays at different medio-lateral and rostral-caudal positions) will likely generate a more complete functional map of the primate amygdala, and (2) these new physiological features revealed by GED can then be set in register with the known neuro-anatomy encompassing all nuclei.

Indeed, addressing these implications through multi-array recordings would ameliorate several of the limitations of the experiments presented here. For example, the use of a single electrode array limited us to assessing network structure along a single axis (i.e., dorsal-ventral) that was dependent on the angle of insertion of the probe. By collecting LFP from contacts on additional arrays placed at different medial-lateral and anterior-posterior positions, component maps could be constructed in multiple planes that intersect the nuclei of the amygdala. These experiments will better localize the sources of the LFP signals and provide a platform for testing anatomy-based hypotheses regarding non-contiguous (i.e., spatially distributed) inter-nuclear subnetworks (*Pitkänen et al., 1995*; *Pitkänen and Amaral, 1991*; *Sah et al., 2003*).

Recent studies have demonstrated inter-nuclear differences in decoding accuracy obtained from pseudo-populations of neurons for various task-related parameters using nearest-neighbor and support-vector machine classifiers (*Grabenhorst et al., 2019*; *Grabenhorst et al., 2016*). In future experiments, GED analyses could be used to generate components that maximally differentiate signals based on similar task parameters. Comparisons of component mapping to the results obtained from these decoding methodologies would provide complimentary metrics for examining the structure-function relationships of single unit and LFP activity.

The subnetworks identified by GED are functionally relevant because the coactivity patterns elicited by visual, tactile, and auditory stimuli discriminated between sensory modalities (as shown in *Figure 4*). While no activity pattern was specific for a particular sensory modality, the majority of activity patterns carried information about one or multiple sensory modalities. For example, the activity of the subnetwork shown in *Figure 4e* conveys information about all sensory modalities albeit with different power in each frequency band, without any one frequency band being assigned to a single sensory modality. Modality-specific analyses further support the hypothesis that functional subnetworks in the amygdala process multisensory information (*Figure 5*). These results complement the selectivity and specificity pattern of the single-unit responses recorded simultaneously from the same contacts (*Morrow et al., 2019*).

There were, however, two notable areas where the components were sensitive to sensory modality: (1) Visual stimuli elicited larger increases in low-frequency (<10 Hz) power (*Figure 4g–i*); and (2)

only the visual components significantly mapped onto MRI-defined nuclear boundaries. This is consistent with the outcome of anatomical tract tracing studies that compared inputs to the primate amygdala from visual, auditory, tactile, and multisensory areas of the temporal lobe and found a preponderance of projections to the amygdala from visual areas (*Amaral and Price, 1984*; *Amaral et al., 1992*; *Stefanacci and Amaral, 2002*). Thus, the mesoscopic intranuclear spatial structure of the multisensory subnetworks is driven predominantly by visual processing. Interestingly, the single-unit data collected in these experiments (*Morrow et al., 2019*) did not show this bias towards the visual modality. This shows that the LFP contains information that was not readily accessible from the spiking data alone. Without monitoring eye movements or facial movements, we can neither confirm nor deny alternative explanations, such as stimulus-related motor responses that could account, at least in part, for the observed pattern of activity. Indeed, neurons in the amygdala respond with phasic bursts of activity to fixations on certain components of images such as faces (*Minxha et al., 2017*) and eyes (*Mosher et al., 2014*), and during production of facial expressions (*Livneh et al., 2012*; *Mosher et al., 2016*).

Further progress in understanding the mesoscale organization of the primate amygdala will come from expanding the network analyses presented here to cover field-field or spike-field interactions between the amygdala and connected structures. The first few attempts in this direction have been successful in revealing the directionality of interactions between the amygdala and the anterior cingulate cortex during aversive learning (*Taub et al., 2018*). Brain-wide circuits are indeed the domain where LFP analyses might be most revealing (*Pesaran et al., 2018*). Brain states, like affect or attention, can be characterized by spatiotemporal interactions between a hub and the cortical areas functionally linked to the hub. For example, the pulvinar and amygdala are hubs placed at the intersection of multiple brain-wide circuits and they both are expected to coordinate the activity of multiple cortical and subcortical areas during behavior (*Bridge et al., 2016*; *Pessoa et al., 2019*). The coordination of spatial attention across brain-wide networks has been recently attributed to directionally selective, theta-band interactions between the pulvinar, the frontal eye fields, and the parietal cortex (*Fiebelkorn et al., 2019*). The multivariate signal decomposition techniques used here, expanded to datasets recorded simultaneously from multiple nodes of amygdala-centered circuits, have great potential to determine how the amygdala coordinates the activity of other structures during social and affective behaviors.

## Materials and methods

### Surgical procedures

Two adult male rhesus macaques, F and B (weight 9 and 14 kg; age 9 and 8 years respectively), were prepared for neurophysiological recordings from the amygdala. The stereotaxic coordinates of the right amygdala in each animal were determined based on high-resolution 3T structural magnetic resonance imaging (MRI) scans (isotropic voxel size = 0.5 mm for monkey F and 0.55 mm for monkey B). A square (26 × 26 mm inner dimensions) polyether ether ketone (a.k.a. PEEK), MRI compatible recording chamber was surgically attached to the skull and a craniotomy was made within the chamber. The implant also included three titanium posts, used to attach the implant to a ring that was locked into a head fixation system. Between recording sessions, the craniotomy was sealed with a silicone elastomer that can prevent growth and scarring of the dura (*Spitler and Gothard, 2008*). All procedures comply with the NIH guidelines for the use of non-human primates in research and have been approved by The University of Arizona's Institutional Animal Care and Use Committee.

### Experimental design

#### Electrophysiological procedures

Local field potential activity was recorded with linear electrode arrays (V-probes, Plexon Inc, Dallas, TX) that have 16 equidistant contacts along a 236 µm diameter shaft. Data were collected using a Plexon OmniPlex data acquisition hardware and software (RRID:SCR_014803). A single electrode array was acutely lowered into the right amygdala for each recording session using a Thomas Recording Motorized Electrode Manipulator (Thomas Recording GmbH, Giessen, Germany). The first contact of the array was located 300 µm from the tip of the probe and each subsequent contact was spaced 400 µm apart; this arrangement allowed us to monitor simultaneously the entire dorso-

ventral expanse of the amygdala. Impedance for each contact typically ranged from 0.2 to 1.2 MΩ. The anatomical location of each electrode was calculated by drawing the chamber to scale on a series of coronal MR images and aligning the chamber to fiducial markers (co-axial columns of high contrast material). Histological verification of these recording site estimates was done in monkey B (see supplemental Materials and methods, and *Figure 3—figure supplement 1*). During recordings, slip-fitting grids with 1 mm distance between cannula guide holes were placed in the chamber, this allowed a systematic sampling of most medio-lateral and anterior-posterior locations in the amygdala. A twenty-three-gauge cannula was inserted through the guide holes and placed 4–6 mm into the cortex. V-probes were driven through the cannula and down to the amygdala at a rate of 70–100 μm per second, slowing to 5–30 μm per second after the tip of the V-probe crossed into the estimated location of the central nucleus. Data from a total of 41 recording sessions monkey (F = 25, monkey B = 16) were analyzed. The raw LFP data used in these analyses is available at https://doi.org/10.5281/zenodo.3752137.

The analog signal from each channel on the V-probe was digitized at the headstage (Plexon Inc, HST/16D Gen2) before being sent through a Plexon pre-amplifier, filtering from 0.1 to 300 Hz and sampling continuously at 40 kHz. LFP was extracted from each contact and down sampled at 1 kHz for analysis. Signals were initially referenced to the shaft of the electrode and were re-referenced offline to the average signal across all electrodes on the probe (i.e., common average reference). This referencing scheme minimizes the contribution of volume conduction from distant sources that spread to all contacts simultaneously and ensured that the recorded LFPs reflected local mesoscale brain dynamics.

## Stimulus delivery

The monkey was seated in a primate chair and placed in a recording booth featuring a 1280 × 720 resolution monitor (ASUSTek Computer Inc, Beitou, Taiwan), two Audix PH5-VS powered speakers (Audix Corporation, Wilsonville, OR) to either side of the monitor, a custom made airflow delivery apparatus (Crist Instruments Company Inc, Damascus, MD), and a juice spout. Juice delivery was controlled by a peristaltic pump (New Era Pump Systems, Inc, Farmingdale, NY, model: NE-9000). The airflow system was designed to deliver gentle, non-aversive airflow stimuli to various locations on the face and head (i.e., the pressure of the air flow was set to be perceptible but not aversive). The system, based on the designs of Huang and Sereno and Goldring et al. (*Goldring et al., 2014*; *Huang and Sereno, 2007*), consists of a solenoid manifold and an airflow regulator (Crist Instruments Company, Inc), which controlled the intensity of the airflow directed toward the monkey. Low pressure vinyl tubing lines (ID 1/8 inch) were attached to ten individual computer-controlled solenoid valves and fed through a series of Loc-line hoses (Lockwood Products Inc, Lake Oswego, OR). The Loc-line hoses were placed such that they did not move during stimulus delivery and were out of the monkey's line of sight. All airflow nozzles were placed ~2 cm from the monkey's fur and outflow was regulated to 20 psi. At this pressure and distance, the air flow caused a visible deflection of the monkey's fur.

Stimulus delivery was controlled using custom written code in Presentation Software (Neurobehavioral Systems, Inc, Berkeley, CA). The monkey's eye movements were tracked by an infrared eye tracker (ISCAN Inc, Burlington, MA, camera type: RK826PCI-01) with a sampling rate of 120 Hz. Eye position was calibrated prior to every session using a 5-point test. During the task the animal was required to fixate for 125–150 ms a central cue ('fixspot') that subtended 0.35 dva. After successful fixation, the fixspot was removed and the monkeys were free to move their eyes around the screen. Removal of the fixspot was followed by the delivery of a stimulus randomly drawn from a pool of neutral visual, tactile, and auditory stimuli. In monkey F, there was no delay between the fixspot removal and stimulus onset, while in monkey B a 200 ms delay was used. Stimulus delivery lasted for 1 s and was followed (after a delay of 700–1200 ms) by juice reward. Each stimulus was presented 12–20 times and was followed by the same amount of juice (~1 mL). Trials were separated by a 3–4 s inter-trial interval (ITI).

For each recording session, a set of eight novel images were selected at random from a large pool of pictures of fractals and objects. Images were displayed centrally on the monitor and covered ~10.5×10.5 dva area. During trials with visual stimuli, the monkey was required to keep his

eye within the boundary of the image. If the monkey looked outside of the image boundary, the trial was terminated without reward and repeated following an ITI.

Tactile stimulation was delivered to eight sites on the face and head: the lower muzzle, upper muzzle, brow, and above the ears on both sides of the head (see *Morrow et al., 2019* for further details). The face was chosen because in a previous study a large proportion of neurons in the amygdala respond to tactile stimulation of the face (*Mosher et al., 2016*). Two 'sham' nozzles were directed away from the monkey on either side of the head to control for the noise made by the solenoid opening and/or by the movement of air through the nozzle. Pre-experiment checks ensured that the airflow was perceptible (caused visible deflection of hair) but not aversive. The monkeys displayed slight behavioral responses (e.g., minor startle responses) to the stimuli during the first habituation session, but they did not overtly respond to these stimuli during the experimental sessions.

For each recording session, a set of eight novel auditory stimuli were taken from freesound.org, edited to be 1 s in duration, and amplified to have the same maximal volume using Audacity sound editing software (Audacity version 2.1.2, RRID:SCR_007198). Sounds included musical notes from a variety of instruments, synthesized sounds, and real-world sounds (e.g., tearing paper). The auditory stimuli for each session were drawn at random from a stimulus pool using a MATLAB script (The MathWorks Inc, Natick, MA, version 2016b, RRID:SCR_001622).

All stimuli were specifically chosen to be unfamiliar and devoid of any inherent or learned significance for the animal. Stimuli with socially salient content like faces or vocalizations were avoided as were images or sounds associated with food (e.g., pictures of fruit or the sound of the feed bin opening). Airflow nozzles were never directed toward the eyes or into the ears to avoid potentially aversive stimulation of these sensitive areas.

## Data analysis

LFP signals were extracted for analysis in MATLAB using scripts from the Plexon MATLAB software development kit and down sampled to 1000 Hz in MATLAB. Artifacts (e.g., signals from broken contacts, sharp spikes in the LFP caused by movement of the animal, or 60 cycle line noise) were removed from the signal prior to further analyses. These data were originally collected for a study designed to assess single-unit processing in the amygdala (*Morrow et al., 2019*). While the number of sessions and the number of trials per session were originally selected with single-unit processing in mind, these experiments provided ample LFP data for analysis as all functioning contacts provided usable LFP data regardless of whether well-isolated single units were present.

All analyses were performed using custom made MATLAB scripts.

### Peri-event LFP

The LFP signal from every trial was taken for each contact for two time windows: a baseline window from −1.5 to −1.0 s relative to fixspot onset and a stimulus delivery window from 0 to +1.0 s relative to stimulus onset. The medians of these two distributions of values were compared using a Wilcoxon rank-sum test to determine the number of contacts with significant event-related changes in LFP signals. These tests were Bonferroni-corrected for multiple comparisons within each session (i.e., tests for differences on each of 16 contacts results in an adjusted alpha level of 0.01/16 = 0.000625).

### Covariance matrices

Covariance matrices were generated by taking the LFP signal for each contact during a time window (baseline or stimulus delivery) to get a timepoints-by-contacts matrix for a given trial. The LFP at each timepoint on each trial was subtracted from the mean LFP across time on the trial. The timepoints-by-contacts matrix was then multiplied by its transpose to create a square, symmetric covariance matrix for a single trial (i.e., contacts-by-contacts covariance matrix). These trial-wise covariance matrices were generated for both the baseline and stimulus delivery time periods. The average of the stimulus covariance matrices made the **S** matrix and the average of the baseline matrices made the **R** matrix.

### Generalized eigendecomposition

Generalized eigendecomposition (GED) was used to generate components that maximized stimulus related changes in activity. GED is a *guided source-separation* technique based on decades of

statistics and engineering work (*de Cheveigné and Parra, 2014*; *Parra et al., 2005*; *Tomé, 2006*; *Van Veen et al., 1997*). Eigendecomposition can be used to decompose a multivariate signal to generate 'components' that capture patterns of covariance across recording contacts. We use generalized eigendecomposition as an optimization algorithm to design a spatial filter (a set of weights across all contacts) that maximizes the ratio of the stimulus covariance matrix to the pre-stimulus covariance matrix (also called the baseline or reference matrix). This can be expressed through the Rayleigh quotient, which identifies a vector $w$ that maximizes the 'multivariate signal-to-noise ratio' between two covariance matrices:

$$w_{max} = argmax\left\{\frac{w^T S w}{w^T R w}\right\}$$

Where $S$ is the covariance matrix generated from data collected during the stimulus delivery time window, $R$ is a covariance matrix generated from the data collected during the baseline window, and $w$ is an eigenvector ($w^T$ is the transpose of $w$). When $w = w_{max}$, the value of the ratio is an eigenvalue, $\lambda$. The full solution to this equation is obtained from a generalized eigendecomposition ($RW\Lambda = SW$, **Figure 2**), where $W$ is a matrix with eigenvectors in the columns, and $\Lambda$ is a diagonal matrix containing the eigenvalues.

The upshot of the generalized eigendecomposition is that the eigenvectors are the directions of covariance patterns that maximally separate the $S$ and $R$ matrices (i.e., contact-by-contact contributions to the component) and eigenvalues contain the magnitude of the ratio between $S$ and $R$ along direction $w$. The goal of this maximization function is therefore to find multichannel covariance patterns that are prominent during stimulus delivery but not during baseline. Should the co-activity patterns be similar during the baseline and stimulus windows, the ratio $S/R$ will be close to one ($\lambda = 1$), however, large differences in multichannel activity that arise during stimulus delivery will manifest as relatively larger eigenvalues ($\lambda \gg 1$).

Note that there are no anatomical or spatial constraints on the decomposition, nor are there any spatial smoothness or peakedness constraints. This means that any interpretable spatial structure that arises from the components results from the nature of the correlations in the data, and not from biases imposed on the analysis method.

To determine a statistical threshold for $\lambda$, we shuffled the labels for the $S$ and $R$ matrices for each trial and performed GED 500 times to create a null distribution of eigenvalues that are associated with activity that was not time-locked to stimulus onset. The observed eigenvalues were compared to this null distribution and components associated with eigenvalues above the 99[th] percentile of this distribution were considered to be statistically significant (similar in concept to a 1-tailed t-test with an alpha level of 0.01).

GED is one of several multivariate decomposition methods that have been explored in neuroscience (others include principal components analysis, independent components analysis, Tucker decomposition, and non-negative matrix factorization). Different methods have different maximization criteria and thus can produce different results. GED has several advantages, including that it is amenable to inferential statistical thresholding, whereas other decompositions are descriptive and thus selecting components for subsequent interrogation may be subjective or biased. Furthermore, validation studies have shown that GED has higher accuracy for recovering ground-truth simulations compared to PCA or ICA (*Cohen, 2017b*; *Haufe et al., 2014a*). Nonetheless, it is possible that different analysis methods can reveal patterns in the data that are not captured by GED.

## Component maps and time series

While eigenvectors are difficult to interpret on their own because they both boost signal and suppress noise (i.e., any irrelevant activity), multiplication of an eigenvector by a covariance matrix creates a forward model of the spatial filter that is easier to visually inspect (*Haufe et al., 2014b*). We refer to these filters as 'component maps' because they convey the relative contribution of each contact to a component signal (i.e., each element of the eigenvector is related to how the LFP on a specific contact contributed to the extracted component). For these data, component maps are generated by multiplying the stimulus covariance matrix, $S$, with the eigenvector, $w$, corresponding to the $n$th component ($Sw_n$).

Component time series were created by multiplying the transpose of the eigenvector matrix by the contacts-by-timepoints LFP data matrix. For example in *Figure 2e*, we show the product of multiplying the first column of the eigenvector matrix (i.e., the eigenvector, $w_1$, associated with the largest eigenvalue) with the matrix of LFP voltage values (labelled as *X*). The first dimension of *X* is dependent on the number of contacts used (e.g., 16) and the second dimension is dependent on the number of timepoints. In this example, the transpose of $w_1$ is a 1-by-16 matrix containing the elements of the eigenvector along the second dimension and *X* is a 16-by-number-of-timepoints matrix. The product of this matrix multiplication is a 1-by-number-of-timepoints matrix that is the component time-series. These time-series data are the weighted average of the activity across contacts that is captured by the component.

## Anatomical grouping versus statistical clustering of contacts

If there is an influence of the spatial location of the contacts on component activity, this should manifest as abrupt shifts in sequential values in the component maps. If these shifts correlate with anatomically defined nuclear boundaries, this would suggest that the cytoarchitectural heterogeneity of the nuclei manifests as a (potentially) functional signal. To assess this possibility, we used the MATLAB function '*findchangepnts*' to identify transitions in the channel weight values. This function works to find the points at which sequential values deviate from some statistical parameter by exhaustively grouping values in all possible sequential configurations and determining the residual error given by a test statistic. We set this function to detect an unspecified number of transitions in the mean values of sequential points (i.e., the '*statistic*' input set to *mean* with no specified minimum or maximum number of transitions). To prevent overfitting, a proportionality constant of 0.05 was used ('*MinThreshold*' input set to 0.05). The proportionality constant is a fixed penalty for adding subsequent changepoints such that new changepoints that do not reduce the residual error by at least 0.05 are rejected (see MATLAB documentation for further details on the findchangepnts function).

To determine if these statistically defined changepoints matched anatomical boundaries estimated via high-resolution MRI, we grouped all contacts on a recording session according to their estimated anatomical location. If all contacts located within a single nucleus were statistically grouped together (i.e., no within nucleus changepoint was detected), each contact was considered to be matching between the two grouping methods. If contacts from multiple nuclei were grouped together, each additional contact from a non-matching nucleus was not included from the total matching count. Likewise, when contacts were grouped anatomically but separated in the statistical clustering, only the grouping that captured the most contacts was counted as matching (e.g., if seven contacts were grouped anatomically but this was split by *changepoints* into one group of 4 contacts and another of 3 contacts, only the 4-contact group was counted as matching). Contacts estimated to be within 200 μm of a nuclear boundary were excluded from this assessment (i.e., not considered matching or non-matching) due to the slight uncertainty in their anatomical grouping. As we were only interested in the spatial information within the amygdala, non-amygdala contacts were also excluded from this analysis.

To determine if the percentage of matching contacts was statistically better than chance, we compared the number of matching contacts obtained from the methods described above to values obtained using cut-and-shift based permutation testing for each significant component. In this method, components maps would be cut at a random point along the 16 element vector and the values before this cut point were shifted to the end of the vector. This new map was then compared to the anatomy-based map (i.e., the anatomy maps are kept as is while the component maps were shuffled). We repeated this 1000 times for each component to get a null distribution of 'matching' values. We then compared the observed distributions of matching values for each component group based on the relative strength of the components (i.e., 1 to 5) to this null distribution using paired t-tests, Bonferroni corrected for the five comparisons ($\alpha$=0.05/5 = 0.01). This allowed us to determine whether the statistical mapping of the average 1st to 5th component (in rank-order) matched the anatomical mapping better than expected by chance.

### Time-frequency decompositions

Component time series were created by multiplying the LFP data with an eigenvector. Time frequency (TF) analysis on the component time series was implemented via convolution with a series of complex Morlet wavelets (logarithmically spaced from 1 to 100 Hz in 80 steps) following standard procedures (*Cohen, 2014*). To determine whether any sections of the time-frequency power differed significantly from baseline we used a cluster-mass test similar in design to the method described in *Maris and Oostenveld, 2007*. TF power matrices were generated for a baseline time window from −1.5 s to −1.0 s relative to fixspot onset and a stimulus time window from 0 to +1.0 s relative to stimulus onset. The difference between these two matrices was then taken to find how TF power changed from baseline to stimulus onset. We then repeated this process shuffling both the labels for the baseline and stimulus time windows and the exact start time of the windows (randomly within 500 ms of actual start time). This generated a set of difference maps that were not time-locked to any specific event during the trials. We repeated this 1000 times and used the mean and standard deviation from these shuffles to z-score normalize the values in the observed difference matrix. Each of the individual difference maps was z-scored using the same parameters as for the observed data. Lastly, clusters of values in these matrices in which the z-scores on adjacent positions were greater than 2.33 (corresponding to $\alpha = 0.01$) were created. The z-scores within these clusters were then summed to create a 'cluster-mass value' for each cluster in the shuffled permutations and the actual data. The cluster-mass values from the difference matrix generated from the observed data was compared to the distribution of maximum cluster-mass values generated by the shuffled permutations (i.e., we compared only the largest cluster in each shuffle to the observed data). Observed clusters-mass values greater than 99% of the values obtained via the shuffles were considered to be statistically significant and are outlined in black in the plots in *Figure 4a–f*.

### Spectral profiles

Power spectra were created from each of the component time series. Principal components analysis was used to extract the prominent features of these spectral profiles across all components. The signals associated with the four principal components accounting for the most explained variance (*Figure 4i–j*, scree plots) were extracted and plotted (*Figure 4i–j*).

### Modality-specific analyses

Modality-specific analyses were conducted by grouping all trials from only one particular sensory domain and re-running the generalized eigendecompositions using the same parameters detailed in the Generalized eigendecomposition section above. Likewise, generation of component maps and the comparisons of the component maps to the MRI-defined nuclear boundaries followed the methods as previously stated in the above sections .

## MRI-based estimations and histological verification of recording sites

In order to verify the accuracy of our recording system, we used a combination of high-resolution MRI and histology. Initially, in vivo 3T MRI scans were performed as described in the main Methods section to guide placement of the V-probe on each recording session. After all data collection from monkey B had finished, we selected a site in the center of the amygdala as the target of an injection of a cell staining dye (Blue Tissue Marking Dye, Triangle Biomedical Sciences, Inc, Durham, NC) to determine the accuracy of estimates of the electrode positions (*Figure 3—figure supplement 1a*). Lastly, ex vivo 7T MRIs and histology were used to verify injection site location.

### Injection procedure

The same methods described for guiding the placement of the electrode arrays during electrophysiological recordings were used to guide the insertion of a 30-gauge cannula into the amygdala of monkey B. A piece of metal tubing (outer diameter 640 µm) was attached near the top of the injection cannula using Gorilla Glue (Maine Wood Concept Inc, Cincinnati, OH) so that the cannula could be placed into the Motorized Electrode Manipulator in the same way as the V-probes. The cannula was thin enough to fit through the same guide tubes used to deliver the V-probes and was lowered at the same rate into the amygdala. A Hamilton syringe, attached to thin rubber tubing, was used to deliver 2 µl of dye over the course of 12 min to ensure ample staining of the target area. We allowed

the cannula to sit for 30 min post injection before removing it from the amygdala. Approximately two hours following dye injection, the animal was prepped for euthanasia, perfused with 4% PFA, and decapitated. The head was placed in 4% PFA for two weeks.

## 7T ex vivo MRI

Fifteen days after euthanasia, the head was scanned using a 7T MRI scan with 250 μm isotropic voxels. Scan parameters were as follows: 86 mm ID quad volume coil; 3D gradient-echo with 250 μm isotropic resolution and matrix size of 320 × 288 × 256; TR = 100 ms; TE = 6 ms; FA = 30; and NA = 6.

## Assessment of accuracy using MRIs and histology

Estimates of the location of the injection site were made independently for each of the three images (3T, 7T, and histology). The images from the 3T, 7T, and histology were then co-registered using major anatomical landmarks. The distances between all injection site estimates were measured and the largest difference in these values was used to ensure the most conservative assessment of the accuracy.

# Acknowledgements

Supported by P50MH100023 and 1R01MH121009 (KMG).

# Additional information

### Funding

| Funder | Grant reference number | Author |
|---|---|---|
| National Institute of Mental Health | P50MH100023 | Katalin M Gothard |
| National Institute of Mental Health | R01MH121009 | Katalin M Gothard |
| European Research Council | StG 638589 | Michael X Cohen |

The funders had no role in study design, data collection and interpretation, or the decision to submit the work for publication.

### Author contributions

Jeremiah K Morrow, Conceptualization, Data curation, Software, Formal analysis, Validation, Investigation, Visualization, Methodology, Writing - original draft, Project administration, Writing - review and editing; Michael X Cohen, Conceptualization, Resources, Data curation, Software, Formal analysis, Supervision, Validation, Visualization, Methodology, Writing - original draft, Project administration, Writing - review and editing; Katalin M Gothard, Conceptualization, Resources, Supervision, Funding acquisition, Validation, Investigation, Visualization, Methodology, Writing - original draft, Project administration, Writing - review and editing

### Author ORCIDs

Jeremiah K Morrow https://orcid.org/0000-0003-0712-6733
Michael X Cohen https://orcid.org/0000-0002-1879-3593
Katalin M Gothard https://orcid.org/0000-0001-9642-2985

### Ethics

Animal experimentation: All procedures comply with the NIH guidelines for the use of non-human primates in research as outlined in the Guide for the Care and Use of Laboratory Animals and have been approved by the Institutional Animal Care and Use Committee of the University of Arizona (protocol #08-101).

Decision letter and Author response
Decision letter https://doi.org/10.7554/eLife.57341.sa1
Author response https://doi.org/10.7554/eLife.57341.sa2

## Additional files

### Supplementary files

- Source code 1. Main GED analysis code.
- Source code 2. Change-point detection code.
- Source code 3. power comparisons code.
- Source code 4. Correlate maps between modality specific GED analyses.
- Source code 5. Function used with *Source code 4* for extracting components of interest.
- Transparent reporting form

### Data availability

All source data (i.e., the raw LFP from all recording sessions) have been deposited in the Zenodo repository (https://doi.org/10.5281/zenodo.3752137). The MATLAB scripts and supporting Excel data files used to process the data shown in each figure are provided with this submission.

The following dataset was generated:

| Author(s) | Year | Dataset title | Dataset URL | Database and Identifier |
|---|---|---|---|---|
| Morrow JK, Cohen MX, Gothard KM | 2019 | Raw LFP data from Gothard Lab Multisensory processing Project | https://doi.org/10.5281/zenodo.3752137 | Zenodo, 10.5281/zenodo.3752137 |

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
