## [Decision Letter]

**Acceptance summary:**

Local field potential activity recorded from the primate amygdala in response to visual, somatosensory, and auditory stimuli was analyzed using the generalized eigendecomposition to identify the change points in the component maps. The change points identified with this novel method corresponded to the known anatomical boundaries between the subdivisions of the amygdala.

**Decision letter after peer review:**

Thank you for submitting your article "Mesoscopic-scale functional networks in the primate amygdala" for consideration by *eLife*. Your article has been reviewed by three peer reviewers, including Daeyeol as the Reviewing Editor and Reviewer #1, and the evaluation has been overseen by a Reviewing Editor and Kate Wassum as the Senior Editor.

The reviewers have discussed the reviews with one another and the Reviewing Editor has drafted this decision to help you prepare a revised submission.

Summary:

The authors analyzed the local field potential activity recorded from the primate amygdala in response to visual, somatosensory, and auditory stimuli. They used a method called the generalized eigendecomposition (GED) to identify the channels in the linear array of electrodes that showed co-variation in their response to sensory stimuli different from the baseline activity, and identified the "change points" in the component maps (i.e., covariance matrix multiplied by an eigenvector). The main finding in this manuscript is that these change points in the component maps roughly and significantly correspond to the known anatomical boundaries between the subdivisions of the amygdala, suggesting that the GED can successfully identify different functional subnetworks within the amygdala. The authors also analyzed the component time series, which is the average LFP weighted by the component, and showed that these component time series can distinguish among different sensory modalities. Although a novel application of the GED can reveal a potentially important principle in the anatomical organization in the amygdala, this was not demonstrated convincingly in this study yet. This may be possible with some additional analyses.

Essential revisions:

1) Although the authors have applied an interesting method to identify LFP activity recorded from multiple channels that covary and carry information about the modality of sensory stimuli, the manuscript does not provide any new insight about the anatomical or functional organization in the amygdala. The way in which GED component weights map onto anatomical boundaries of amygdala nuclei (Figure 3) is impressive, and these results are strengthened by the careful histological and MRI-based reconstruction of recording sites. Nevertheless, this seems merely a validation of the method. Would it be possible to discover novel principle, for example, by applying GED separately for different sensory modalities? Similarly, if GED is applied to maximally distinguish between the LFP recorded in response to different sensory modalities, will the resulting components different from those identified in the current study?

More specifically, each session had 8 visual, 8 tactile, and 8 auditory stimuli, and the anatomically-related functional network analysis was done by averaging across the covariance matrices corresponding to all these types of stimuli. How do the functional maps look if the effect of each stimulus type was investigated separately? It would be interesting to see how component map separates among the amygdala nuclei when different sensory modalities are not averaged in. When we talk about functional networks, the 'function' itself is quite important in understanding the contribution of the network. It will be important to test if the function-anatomy boundaries are same or different if they show the functional separations for the visual, tactile, and auditory stimuli separately. This is largely because different sub-regions might be contributing differently depending on the content of the information. For this reason, if the boundaries remain similar, that would strengthen the finding.

2) The authors state that LFPs allow for a novel way of clustering functional network within the amygdala sub-regions in a manner that cannot be resolved from single-unit activity. However, it would be even more convincing and more direct if the authors could pit these results (i.e., ability to identify functional network) directly with single-unit / multi-unit data collected from the same contact sites in this same paper. In particular, it would be useful to show the multi-unit data as a comparison since the resolution becomes lower than single units but still are in the spike domain. Since the authors have both of these data (Morrow, Mosher and Gothard, 2019), it might be powerful if they could directly compare the LFP-based metrics to spike-based metrics to inspect whether the functional separability reported here is only present (or is better). This will be very helpful even though their previous work showed poor or no functional separability of the amygdala nuclei based on single-unit activity.

3) Another weakness in this study is that the animal's behavior was not well controlled during the baseline period. This 1-s baseline period includes the onset of fixation target and eye movements to fixate that target. Therefore, any changes in LFP associated with these two events could be mistaken for the activity related to stimulus presentation. The fact that the component time series are different for different sensory modalities mitigate this problem to some degree, but the eye movement data should be analyzed more carefully. For example, were there differences between stimulus conditions in the rate with which the animals broke fixation? Such effects could indicate differences in motivation between the conditions which would complicate an interpretation of the findings in Figure 4 purely in terms of sensory modalities.

4) The mapping of GED component weights to anatomical divisions relies on the detection of change points between consecutive weights. Would the approach be sensitive to detect similarities in weights between non-contiguous subregions of the amygdala? For example, Pitkanen and Amaral, (1998) showed with tracer injections that specific focal areas in the dorsal lateral nucleus label focal areas both in the ventral (non-contiguous) lateral nucleus and in the distant accessory basal nucleus. Such anatomical networks between non-contiguous amygdala areas may well form functional networks, and this could be reflected in co-activity patterns in the present data.

5) It would be helpful to discuss how the present results, and GED results generally, depend on the penetration angle of the probe. For example, there may well be functional networks between lateral and accessory basal nucleus, but these may not be detectable with a vertical penetration that does not sample these nuclei simultaneously. This is not a criticism of the present recordings, but it would be good to discuss what recordings would be required to map amygdala networks comprehensively.

6) When analyzing the time-frequency spectra in the LFP signals corresponding to various stimuli, the authors report that the spectral profiles remain similar even after they have removed the contacts outside the amygdala. However, in some places, there seem to be visible differences and some alterations. For example, the scree plot summary showing the number of significant components in Figure 2H and Figure 2J seem rather different; the component-to-frequency mappings do not seem to be clearly maintained between Figure 4K and 4L; and there are some visible changes in Figure 4—figure supplement 1 as well. It is unclear how to interpret these differences in such cases.

7) The authors may want to acknowledge some human neuroimaging data that have identified, or attempted to identify, sub-region-based amygdala sub-networks. Because such studies also argue for mesoscale organizations of the amygdala, discussing this will increase impact of the finding – it would be informative for the authors to discuss how the current findings inform such fMRI findings in humans in the Discussion section. Given that fMRI signals are "more like" LFP signals, it might be worthwhile to discuss although the human networks identified still do not have the same anatomical resolution as in the current study. Some potentially relevant references include:

Bickart et al., (2012).

Roy et al., (2009).

[Editors' note: further revisions were suggested prior to acceptance, as described below.]

Thank you for resubmitting your work entitled "Mesoscopic-scale functional networks in the primate amygdala" for further consideration by *eLife*. Your revised article has been evaluated by Kate Wassum (Senior Editor) and a Reviewing Editor.

The manuscript has been improved but there are some remaining issues that need to be addressed before acceptance, as outlined below:

1) With respect to the question of whether the GED approach could uncover a novel functional principle in the amygdala, the new analyses in Figure 5 and the related Results text go in that direction, but this is not very clear yet. The results suggest that GED components from visual trials matched nuclear boundaries better than components from auditory and tactile trials (subsection “Modality-specific GED analyses”). The authors derive the attractive conclusion (Discussion section) that "the mesoscopic intranuclear spatial structure of the multisensory subnetworks is driven predominantly by visual processing", consistent with known visual input patterns to amygdala. This could be a key point of the paper but the data that support this conclusion are not clearly shown and merely stated in subsection “Modality-specific GED analyses”. It would be very helpful if the authors actually quantify which modality is better across all the data. They can also include a panel in Figure 5 to clearly document this effect, and perhaps visualize the similarity to the amygdala visual inputs known from tracing studies. This would help the paper to derive new insight about the amygdala's functional organization beyond validating the GED method.

2) Although the authors have made some clarifications, it seems still possible whether differences between sensory conditions in LFP signals could be partly related to different motor responses to the different cues. For example, visual stimuli could provoke different gaze patterns as the animal explores the stimulus (focusing on the screen) compared to auditory stimuli (the animal might look at the sound source). Similarly, the tactile stimulus might evoke specific muscle activations (e.g. face expressions). It is worth mentioning in the discussion that, in addition to the sensory stimuli themselves, some of the reported neural effects could be influenced by stimulus-related motor responses.

3) The authors argue in the introduction that an LFP approach may be needed to uncover functional networks, and that single-neuron approaches "may not carry sufficient information about the state of the network to accurately describe the neural computations taking place therein". However, the present findings do not fully deliver on this promise or justify such statements. For example, in the reviewers' comments, stimulus discrimination was suggested as a test but this did not show very strong results. This contrasts markedly with the often exquisite discriminations afforded by single neurons (indeed, as documented in the authors' own previous papers). Accordingly, it might be better to tone down the statements about functional networks and limitations of single-neuron approaches in Introduction and Discussion section.

4) Subsection "Modality-specific GED analyses": "Furthermore, the sensory-modality-specific component maps were significantly correlated with the component maps that combined all three sensory modalities (Figure 5D-K)." Should this refer to Figure 5C? There is not Figure 5D-K.

5) Figure 4L: the effect of removing non-amygdala sources on spectral profiles are still not very clear. Please include an interpretation of why the highlighted peaks are associated with different components in panels K and L (4, 7, 70 Hz).

---

## [Author Response]

Summary:The authors analyzed the local field potential activity recorded from the primate amygdala in response to visual, somatosensory, and auditory stimuli. […] This may be possible with some additional analyses.

We thank the reviewers for suggesting additional analyses to elevate the findings of this study above a mere validation of the method (which was indeed one of our goals). We have addressed each criticism and revised the manuscript based on the suggestions of the reviewers. We have addressed the concerns of the reviewers as follows:

Essential revisions:1) Although the authors have applied an interesting method to identify LFP activity recorded from multiple channels that covary and carry information about the modality of sensory stimuli, the manuscript does not provide any new insight about the anatomical or functional organization in the amygdala. The way in which GED component weights map onto anatomical boundaries of amygdala nuclei (Figure 3) is impressive, and these results are strengthened by the careful histological and MRI-based reconstruction of recording sites. Nevertheless, this seems merely a validation of the method. Would it be possible to discover novel principle, for example, by applying GED separately for different sensory modalities? Similarly, if GED is applied to maximally distinguish between the LFP recorded in response to different sensory modalities, will the resulting components different from those identified in the current study?More specifically, each session had 8 visual, 8 tactile, and 8 auditory stimuli, and the anatomically-related functional network analysis was done by averaging across the covariance matrices corresponding to all these types of stimuli. How do the functional maps look if the effect of each stimulus type was investigated separately? It would be interesting to see how component map separates among the amygdala nuclei when different sensory modalities are not averaged in. When we talk about functional networks, the 'function' itself is quite important in understanding the contribution of the network. It will be important to test if the function-anatomy boundaries are same or different if they show the functional separations for the visual, tactile, and auditory stimuli separately. This is largely because different sub-regions might be contributing differently depending on the content of the information. For this reason, if the boundaries remain similar, that would strengthen the finding.

In our initial analyses, we pooled trials across modalities for three reasons: (1) pooling data maximized the signal-to-noise ratio, which improved the statistical reliability of the matrix decompositions; (2) modality-independent decompositions ensured that our findings were not biased by data selection; (3) our previous single-unit results show that networks of multisensory neurons are distributed throughout the amygdala.

However, the reviewers bring up an excellent point. The GED analysis can be specifically tailored to maximize modality-specific differences. We therefore ran a series of modality-specific analyses. As reported in detail in the revised manuscript, these findings show that many features of our components analysis remained modality-independent, while some features (e.g., component mapping onto nuclear boundaries) showed some modality-specific patterns. These new analyses enhance the impact of our paper, and we appreciate the suggestion. We added a new Figure 5 to illustrate these results as well as expanded the Results section and Discussion section.

2) The authors state that LFPs allow for a novel way of clustering functional network within the amygdala sub-regions in a manner that cannot be resolved from single-unit activity. However, it would be even more convincing and more direct if the authors could pit these results (i.e., ability to identify functional network) directly with single-unit / multi-unit data collected from the same contact sites in this same paper. In particular, it would be useful to show the multi-unit data as a comparison since the resolution becomes lower than single units but still are in the spike domain. Since the authors have both of these data (Morrow, Mosher and Gothard, 2019), it might be powerful if they could directly compare the LFP-based metrics to spike-based metrics to inspect whether the functional separability reported here is only present (or is better). This will be very helpful even though their previous work showed poor or no functional separability of the amygdala nuclei based on single-unit activity.

This is an important point, and we appreciate the reviewers bringing this up. In our 2019 paper, we showed a lack of regional clustering of neurons with unisensory and various combinations of multisensory responses. It was clear that the state of the network was not adequately captured by the sparsely sampled single units along a single V-probe in the amygdala (typically less than 10 simultaneously monitored neurons).

In fact, we have tried some of the analyses suggested by the reviewers (using the same dataset; presented at SFN last year). To determine whether the coupling between single units and LFP’s is modality-specific, we computed the spike-field coherence and found that the pairwise phase consistency frequently differed between the three sensory modalities, suggesting that stimuli of different sensory modalities may engage independent (though potentially overlapping) networks of neurons. While pairwise phase consistency was prominent at low (~1-10 Hz) and some higher (~40 Hz) frequencies in the single channel data, high frequency pairwise phase consistency was generally attenuated in GED-based components (Morrow et al., 2019). This by itself suggested that GED-based assessments of spike field coherence may be more resistant to high frequency artifacts from multiunit and single unit activity.

Indeed, Belitski et al., (2010) showed that the extraction of stimulus information from frequency bands higher than 50Hz required averaging hundreds of milliseconds of data. This is not possible in our case because, unlike in the motor cortex cortex where neurons show prolonged modulations of firing rate in relation to movement, a large proportion of neurons in the amygdala respond to stimuli with a sharp phasic change of firing rate. This phasic response rarely lasts longer than 150ms. Typically, significant changes of firing are observed 110-140 ms after stimulus onset and the firing rate returns to baseline 250-300 ms from stimulus onset (Gothard et al., 2007; Mosher et al., 2010; Morrow et al., 2019) leaving only about 100 ms of useful multiunit activity for analysis. After this 100 ms window, a large number of phasic neurons drop out from the neural ensemble activated by each stimulus. For these reasons it is unlikely that the approach suggested by the reviewers would add to our report.

We chose not to include these findings here because (1) we realized while working on the poster that a sufficiently rigorous treatment of this question requires more methodological and analysis work, which would detract from this paper, and (2) we felt that we had insufficient number of simultaneously recorded neurons to address this question satisfactorily. In fact, we plan to record, in the immediate future, from 2x32 contacts/amygdala (two V-probes with 32 contacts each placed at different anterior-posterior and medio-lateral locations in the amygdala) that will provide a higher yield of single units and also a better spatial resolution for the anatomical topography of hypothesized subnetworks detailed in the manuscript. The larger number of neurons may compensate for the short duration each neuron is active.

3) Another weakness in this study is that the animal's behavior was not well controlled during the baseline period. This 1-s baseline period includes the onset of fixation target and eye movements to fixate that target. Therefore, any changes in LFP associated with these two events could be mistaken for the activity related to stimulus presentation. The fact that the component time series are different for different sensory modalities mitigate this problem to some degree, but the eye movement data should be analyzed more carefully. For example, were there differences between stimulus conditions in the rate with which the animals broke fixation? Such effects could indicate differences in motivation between the conditions which would complicate an interpretation of the findings in Figure 4 purely in terms of sensory modalities.

The reviewers have caught an important omission in our description of the experiment. When we initially began analyzing these data we used a baseline that was relative to stimulus onset; however, we changed our methods so that the baseline would be generated relative to the time of fixspot onset for each trial in order to avoid the issues that the reviewers flagged here (but we did not adjust the text to reflect this change). We have updated the text to better explain that the behavioral task was designed to separate activity during a window before the pre-stimulus fixation cue from stimulus-related activity.

Further, we required the monkeys to fixate before the delivery of stimuli, partly to have them initiate a trial (motivation, vigilance) and partly to ensure that they attend to the monitor for the presentation of the visual stimuli. We did not maintain the requirement to fixate on a central point once the stimulus was present. The updated text shows that the baseline window does not include the fixation window (which was very short = 125-150 ms, see Results section, Materials and methods section and Figure 1) and that during the presentation of the stimuli the monkey was no longer required to fixate a central point (Materials and methods section). Therefore, the animals were free to move their eyes during both the baseline (pre-fixation) and stimulus delivery windows. Lastly, we shifted the baseline window forward and backwards by 500 ms to further ensure that our results were not dependent on the data in the baseline window. This resulted in very minor changes in the total number of significant components (111 and 119, respectively, both less than 5% change), suggesting that the results we obtained are not attributable to the data used for baseline.

4) The mapping of GED component weights to anatomical divisions relies on the detection of change points between consecutive weights. Would the approach be sensitive to detect similarities in weights between non-contiguous subregions of the amygdala? For example, Pitkanen and Amaral, (1998) showed with tracer injections that specific focal areas in the dorsal lateral nucleus label focal areas both in the ventral (non-contiguous) lateral nucleus and in the distant accessory basal nucleus. Such anatomical networks between non-contiguous amygdala areas may well form functional networks, and this could be reflected in co-activity patterns in the present data.

Yes, the GED method can detect non-continuous networks. In fact, one advantage of GED is that it is completely blind to spatial and anatomical information. It is a purely statistical decomposition that has no spatial/anatomical constraints. This is part of the reason why it’s remarkable that we see a strong convergence with the estimated anatomical nuclear subdivisions — these concordances naturally arise from the data and are not imposed onto the method.

Inspection of Figure 3 shows that several components were driven by trans-nuclear networks (in particular, panels C, E, and F).

We have included additional text the Discussion section and Materials and methods section, to highlight this feature to readers.

5) It would be helpful to discuss how the present results, and GED results generally, depend on the penetration angle of the probe. For example, there may well be functional networks between lateral and accessory basal nucleus, but these may not be detectable with a vertical penetration that does not sample these nuclei simultaneously. This is not a criticism of the present recordings, but it would be good to discuss what recordings would be required to map amygdala networks comprehensively.

Again, an excellent point and something we did not emphasize enough in the discussion. Indeed, expanding GED to three-dimensions (using multiple linear probes placed at different medio-lateral and rostral-caudal positions) will likely generate a more complete functional map of the primate amygdala, one that could potentially track the processing flow from the lateral, to the basal and accessory basal nuclei. These types of experiments are exactly what we hope to execute in the near future. We have expanded the Discussion section that addresses this point.

6) When analyzing the time-frequency spectra in the LFP signals corresponding to various stimuli, the authors report that the spectral profiles remain similar even after they have removed the contacts outside the amygdala. However, in some places, there seem to be visible differences and some alterations. For example, the scree plot summary showing the number of significant components in Figure 2H and Figure 2J seem rather different; the component-to-frequency mappings do not seem to be clearly maintained between Figure 4K and 4L; and there are some visible changes in Figure 4—figure supplement 1 as well. It is unclear how to interpret these differences in such cases.

Thank you for pointing this out. This was confusingly written in the original manuscript. Figure 2 shows that the prominent spectral peaks are preserved when excluding the components that contained non-amygdala sources. In particular, we highlight the peaks at 4, 7, 38, and 70 Hz. On the other hand, the spectral profiles are clearly not identical, which is to be expected — the spectra in panel **k** include contributions from the hippocampus and rhinal cortex. We have now clarified this text in the Discussion section.

7) The authors may want to acknowledge some human neuroimaging data that have identified, or attempted to identify, sub-region-based amygdala sub-networks. Because such studies also argue for mesoscale organizations of the amygdala, discussing this will increase impact of the finding – it would be informative for the authors to discuss how the current findings inform such fMRI findings in humans in the Discussion section. Given that fMRI signals are "more like" LFP signals, it might be worthwhile to discuss although the human networks identified still do not have the same anatomical resolution as in the current study. Some potentially relevant references include:Bickart et al., (2012).Roy et al., (2009).

These are indeed pioneering studies that already alluded to this organization scheme that transcend the nuclear boundaries and thus belong to the Discussion section.

[Editors' note: further revisions were suggested prior to acceptance, as described below.]

The manuscript has been improved but there are some remaining issues that need to be addressed before acceptance, as outlined below:1) With respect to the question of whether the GED approach could uncover a novel functional principle in the amygdala, the new analyses in Figure 5 and the related Results text go in that direction, but this is not very clear yet. The results suggest that GED components from visual trials matched nuclear boundaries better than components from auditory and tactile trials (subsection “Modality-specific GED analyses”). The authors derive the attractive conclusion (Discussion section) that "the mesoscopic intranuclear spatial structure of the multisensory subnetworks is driven predominantly by visual processing", consistent with known visual input patterns to amygdala. This could be a key point of the paper but the data that support this conclusion are not clearly shown and merely stated in subsection “Modality-specific GED analyses”. It would be very helpful if the authors actually quantify which modality is better across all the data. They can also include a panel in Figure 5 to clearly document this effect, and perhaps visualize the similarity to the amygdala visual inputs known from tracing studies. This would help the paper to derive new insight about the amygdala's functional organization beyond validating the GED method.

We have quantified and statistically compared the differences between sensory modalities in terms of how well the computed components predict the estimated nuclear boundaries in the amygdala. Instead of an additional panel to Figure 5, we have added a new table that contains more detailed statistical information than what we could provide in subplot or a graphical representation. This table shows that the first two components calculated for visual stimuli predict reliably the nuclear boundaries (even after conservative Bonferroni corrections). The first component calculated for responses to auditory stimuli approaches significance but was trending in the opposite direction (i.e., showed worse matching than the null distribution generated from randomly shuffled component maps). The components maps calculated for tactile stimuli showed no statistically significant relationship to nuclear boundaries. These results, together with those presented in Figure 4, suggest that the visual inputs carry the lion’s share of modality-dependent activity patterns in the amygdala. This outcome confirms the predictions of multiple anatomical tract tracing studies, namely that the amygdala receives a disproportionately larger volume of anatomical inputs from visual areas. We have described this in the text (subsection “GED components discriminate between sensory modalities”).

2) Although the authors have made some clarifications, it seems still possible whether differences between sensory conditions in LFP signals could be partly related to different motor responses to the different cues. For example, visual stimuli could provoke different gaze patterns as the animal explores the stimulus (focusing on the screen) compared to auditory stimuli (the animal might look at the sound source). Similarly, the tactile stimulus might evoke specific muscle activations (e.g. face expressions). It is worth mentioning in the Discussion section that, in addition to the sensory stimuli themselves, some of the reported neural effects could be influenced by stimulus-related motor responses.

We agree that without monitoring eye movements, the movements of the pinna or of the facial musculature during the presentation of sensory stimuli, we can neither confirm nor deny that stimulus-related motor responses could account for the observed pattern of activity in the amygdala. We have added to the Discussion section addressing this possibility and supported it citations from the literature.

3) The authors argue in the Introduction that an LFP approach may be needed to uncover functional networks, and that single-neuron approaches "may not carry sufficient information about the state of the network to accurately describe the neural computations taking place therein". However, the present findings do not fully deliver on this promise or justify such statements. For example, in the reviewers' comments, stimulus discrimination was suggested as a test but this did not show very strong results. This contrasts markedly with the often exquisite discriminations afforded by single neurons (indeed, as documented in the authors' own previous papers). Accordingly, it might be better to tone down the statements about functional networks and limitations of single-neuron approaches in Introduction and Discussion section.

At the request of the reviewers, we have removed this controversial statement from the Introduction and toned down our rhetoric in the Discussion section.

4) Subsection "Modality-specific GED analyses": "Furthermore, the sensory-modality-specific component maps were significantly correlated with the component maps that combined all three sensory modalities (Figure 5D-K)." Should this refer to Figure 5C? There is not Figure 5D-K.

Thank you for catching this error. Yes, the text should have read 5C and has been corrected.

5) Figure 4L: the effect of removing non-amygdala sources on spectral profiles are still not very clear. Please include an interpretation of why the highlighted peaks are associated with different components in panels K and L (4, 7, 70 Hz).

We agree that this point required more detailed explanations. We have added a paragraph to subsection “GED components discriminate between sensory modalities” that explains differences in the spectral profiles for both cases: including or excluding the electrodes estimated to be outside the amygdala. Given the importance of these observations for comparing the spectral profiles in the amygdala to the cortex, we have updated Figure 4—figure supplement 1 in an attempt to make this point more clear.